



# Wind Reconstruction Algorithm for Viking Lander 1

Tuomas Kynkäänniemi[1], Osku Kemppinen[2,3], Ari-Matti Harri[2], and Walter Schmidt[2]

[1]School of Science, Aalto University, Espoo, Finland
[2]Earth Observation, Finnish Meteorological Institute, Helsinki, Finland
[3]Currently at the Department of Physics, Kansas State University, Manhattan, Kansas, USA

*Correspondence to:* Tuomas Kynkäänniemi (tuomas.kynkaanniemi@fmi.fi)

**Abstract.**

The wind measurement sensors of Viking Lander 1 (VL1) were only fully operational for the initial phase of the mission. We have developed an algorithm for reconstructing the wind measurement data after the wind measurement sensor failures. The algorithm for wind reconstruction enables the processing of wind data during the complete VL1 mission. The heater element of

5 the quadrant sensor, which provided auxiliary measurement for wind direction, failed during the 45th sol of the VL1 mission. Additionally, one of the wind sensors of VL1 broke down during the sol 378. Regardless of the failures, it was still possible to reconstruct the wind measurement data, because the failed components of the sensors did not prevent the determination of the wind direction and speed, as some of the components of the wind measurement setup remained intact for the complete mission.

This article concentrates on presenting the wind reconstruction algorithm and methods for validating the operation of the

10 algorithm. The algorithm enables the reconstruction of wind measurements for the complete VL1 mission. The amount of available sols is extended from 350 to 2245 sols.

## 1 Introduction

The primary goal of the Viking Mission was to investigate the current or past existence of life on Mars. The Viking Lander's payload consisted of instruments designed for meteorological experiments, seismological measurements, and instruments for

experiments on the composition of the atmosphere (Chamberlain et al., 1976; Soffen, 1977).

VL1 landed on Mars on 20.7.1976. The location of the landing spot was a low plain area named Chryse Planitia, which has a slope rising from south to west. The landing coordinates of VL1 were 22° N, 48° W (Soffen, 1977).

VL1 operated for 2245 Martian sols, which is much longer than was expected (Soffen, 1977). Thus the data set produced is significant in size. The VL1 data set enables the study of various meteorological phenomena from diurnal variations to seasonal

variations in temperature, pressure and wind direction, as well as wind speed.

The Finnish Meteorological Institute (FMI) has developed a set of tools that enable processing the Viking Lander meteorological data beyond previously publicly available data (Kemppinen et al., 2013). Currently National Aeronautics and Space Administration's (NASA) Planetary Data System (PDS) contains wind measurement data only from 350 sols. The FMI tools make it possible to process data from full 2245 sols instead, over a six-fold increase.



The state of the wind measurements from the surface of Mars is quite poor, because the wind measurements of other missions to the surface of Mars have either failed or produced a shorter data set than that of VL1. After the Viking Mission there was a twenty year pause before the research on the surface of Mars was finally resumed. The Mars Pathfinder conducted wind measurements on the surface of Mars for 86 sols (Golombek et al., 1999), results of which are presented in (Schofield et al., 1997). The Mars Science Laboratory (MSL) can surpass the extent of the VL1 data set in year 2018, but the wind measurement setup of the MSL is not fully functional due to hardware issues. Thus, the VL1 wind measurement data set remains the longest data set of wind measurements from the surface of Mars at least for the coming years.

This article focuses on the reconstruction of the VL1 wind measurements. The wind reconstruction algorithm is defined in Sec. 3 and the validation methods for the algorithm are presented in Sec. 4.1. The reconstructed data from the VL1 wind measurements is presented in Sec. 5.

The significance of this work lies in the fact that other missions on the surface of Mars have not yet succeeded to measure a data set equal to the size of that of VL1. Even though the VL1 landed on Mars 40 years ago, not all the data from the VL1 wind instruments have been analysed and published due to various complications that are described below.

## 2   Wind measurement setup and sensor malfunctions

### 2.1   Wind sensors

Viking Lander wind measurement setup consisted of two hot-film wind sensors and a quadrant sensor (Chamberlain et al., 1976). The hot-film wind sensors were designed to determine the wind velocity component normal to each of the sensors, and the quadrant sensor was designed to provide information about the wind direction and therefore solve the four-fold ambiguity of the normal components. (Chamberlain et al., 1976).

The wind sensor design is presented in Fig. 1. The hot film wind sensors were mounted at a $90°$ angle with respect to each other and the temperature of the films was maintained at $100°C$ above the ambient gas temperature. The wind velocity normal to each hot-film sensor could be determined from the power required to maintain the overheat temperature against heat loss due to radiation and conduction. Assuming the wind velocity is a vector $\boldsymbol{v}$ from direction $\theta$, as presented in Fig. 1, the perpendicular wind speed components $v_x$ and $v_y$ can be determined by

$$v_y = |\boldsymbol{v}|\sin(\theta) \tag{1}$$

$$v_x = |\boldsymbol{v}|\cos(\theta) \tag{2}$$

There exists a four-fold ambiguity in the wind velocity measured by the two wind sensors. The ambiguity is caused by the wind sensors only measuring the wind velocity component normal to the sensor and it is resolved using the quadrant sensor. (Davey et al., 1973; Sutton et al., 1978)





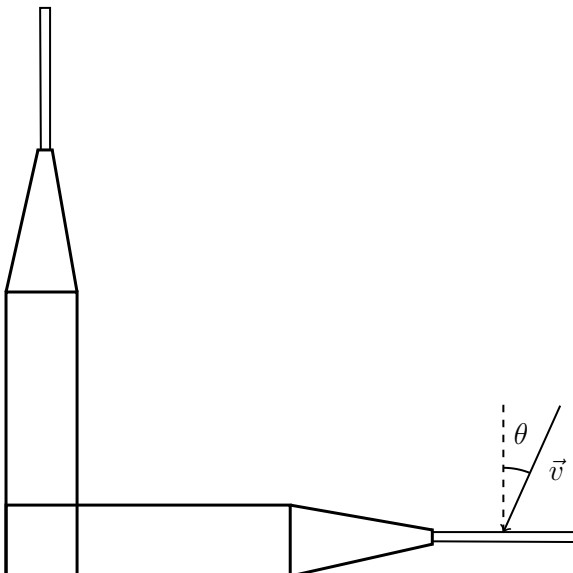

**Figure 1.** Wind sensor design. The hot-films were mounted at the end of two separate holders, which were set perpendicular to each other. (Davey et al., 1973)

## 2.2 Quadrant sensor

The quadrant sensor was designed to provide a secondary measurement to solve the ambiguity in the wind direction. The design of the sensor is presented in Fig. 2. The operating principle of the sensor is based on locating the thermal wake of a heated vertical cylinder. The location of the wake is determined from the temperature distribution about the cylinder using four

5  chromel-constantan thermocouples. These thermocouples were connected in series, and each pair measures the temperature difference across the sensor due to the thermal wake (Davey et al., 1973; Sutton et al., 1978).

## 2.3 SANMET

The data used in the wind reconstruction algorithm is from NASA's Science Analysis of Meteorology (SANMET) program (Buehler, 1974). SANMET calculates the values of various meteorological quantities, such as wind speed, wind direction,

10  temperature and pressure, from the voltage signals of the meteorology instruments. The operating principle and methods used by SANMET program are presented in detail in (Buehler, 1974). The wind reconstruction algorithm requires the voltages $V_{\mathrm{QS}_1}$ and $V_{\mathrm{QS}_2}$ of the quadrant sensor's thermocouples. These voltages can be read from the SANMET output under the DATA6 header. The algorithm additionally requires the wind directions $\theta$ and the Nusselt numbers, $Nu_1$ and $Nu_2$, of the wind sensors during the sols 1-45.





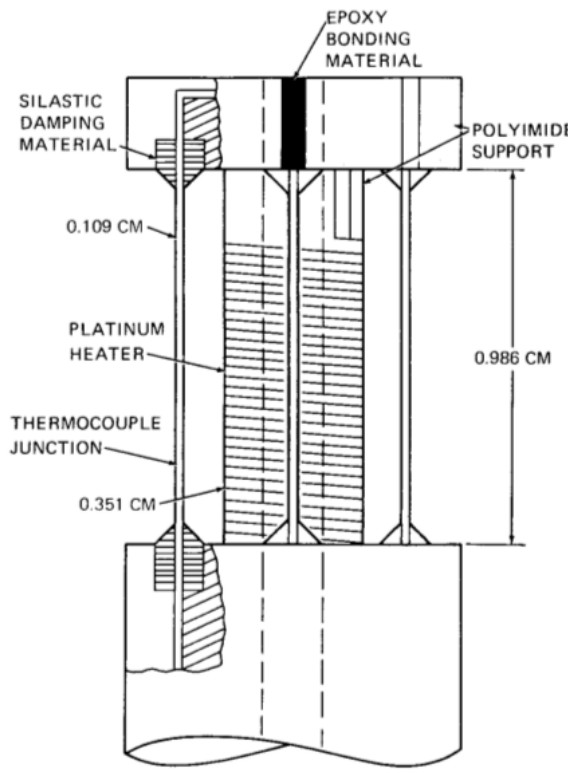

**Figure 2.** Quadrant sensor design. (Hess et al., 1977)

## 2.4   VL1 sensor malfunctions

The heater element of the VL1 quadrant sensor was thought to be damaged during the 45th sol (Murphy et al., 1990; Hess et al., 1977). During the sol 46 there exists a sudden change in the behavior of the voltage values of the thermocouple pairs QS1 and QS2, which is shown in Fig. 3. When the quadrant sensor was functioning in the intended way, the voltage values of QS1 and QS2 varied between ±5.0 mV, but after the failure of the heater element the variation range of the voltages changed to approximately ±0.5 mV.

After the failure of the quadrant sensor's heater element, the SANMET process for determining the wind direction was no longer fully reliable. Thus another method for obtaining the wind direction was required and therefore an algorithm for wind data reconstruction was developed. The algorithm assumes that both of the quadrant sensor thermocouples remained functional for the whole VL1 mission. Therefore the instrument can be used whenever the radiance of the Sun is strong enough to heat the quadrant sensor's heater element to a significantly higher temperature than the ambient temperature.

In addition to the failure of the quadrant sensor's heater element, one of the two hot-film wind sensors of VL1 broke down during sols 377-378. The decay of the VL1 wind sensor 2 is illustrated in Fig. 4 by presenting the unbinned values of the Nusselt



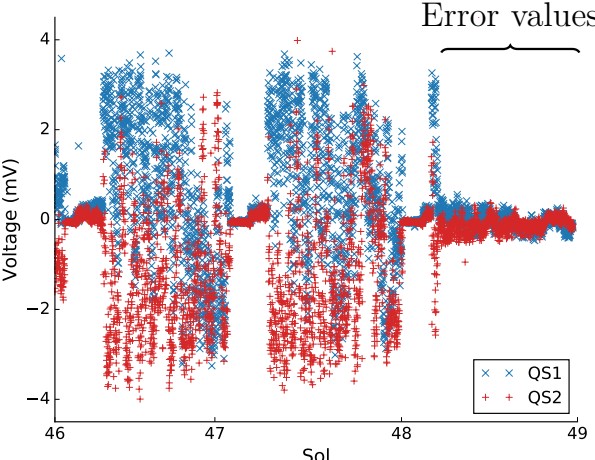

**Figure 3.** Failure of the heater element of VL1's quadrant sensor during sol 46.

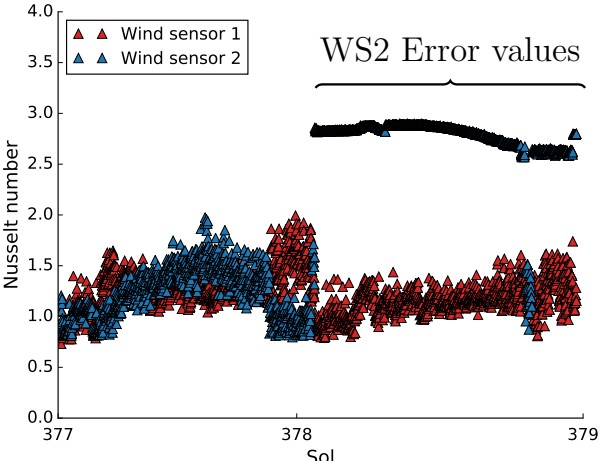

**Figure 4.** The decay of the VL1's wind sensor 2 (WS2) during the sols 377-378.

number measured by both of the wind sensors. At the turn of the sols 377 and 378, the dynamic of the wind sensor 2 changed so that the sensor obtains almost constant Nusselt number of slightly less than 3.0. At the end of sol 378's measurements the wind sensor 2 resumed nominal function for a very short period of time, after which the behavior of the sensor returned to the failure state permanently.





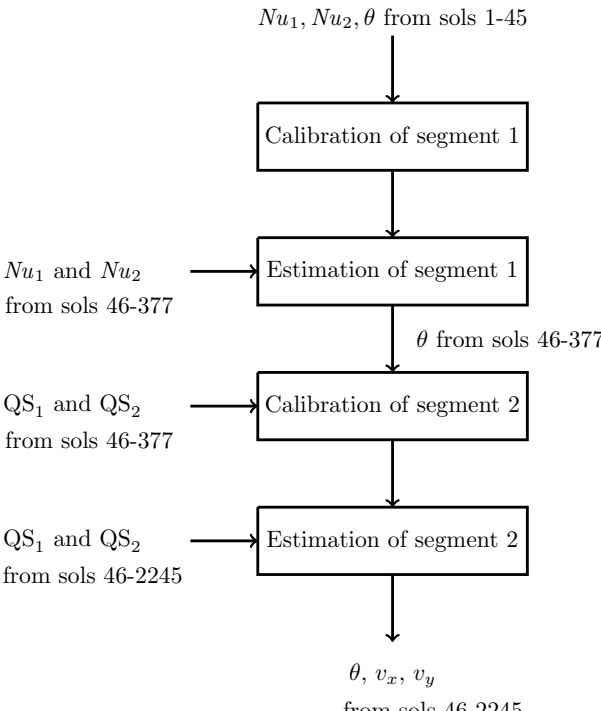

**Figure 5.** Schema of the complete processing chain of the wind reconstruction algorithm. $Nu_1$ and $Nu_2$ are the Nusselt numbers of wind sensors 1 and 2, $QS_1$ and $QS_2$ are the voltage values of the thermocouple pairs, $\theta$ is the wind direction and $v_x$ and $v_y$ are the wind speed normal components.

## 3    Wind reconstruction algorithm

### 3.1    The re-analysing process of the wind measurements

Due to the failures of the quadrant sensor and one of the hot-film wind sensors, the reconstruction of wind directions and speeds have to be carried out in two segments. The complete work-flow of the wind reconstruction algorithm is presented in the schema of Fig. 5. Both of the segments contain at first a calibration phase, where the calibration function is calculated, and an estimation phase, where the wind direction $\theta$ is estimated.

In the first segment the wind directions of the fully functional quadrant sensor are used to calibrate the two hot-film sensors' Nusselt numbers. The method of calibrating the Nusselt numbers with the correct wind direction data from sols 1-45 is presented in Sec. 3.2. The Nusselt numbers of the two hot-film wind sensors are then used for determining the wind direction during sols 46-377. The reconstruction of wind direction after the sol 376 is unfortunately impossible using the calibrated Nusselt numbers, as one of the two sensors failed during sol 378. Therefore in the second segment, the reconstructed wind di-





rections from sols 46-377 are required for calibrating the weak voltage signals of the quadrant sensor, to enable the estimation of wind direction after sol 377 with the quadrant sensor.

In the second segment the reconstructed wind directions, determined using the Nusselt numbers of wind sensors, are used to calibrate the quadrant sensor voltages. The quadrant sensor thermocouple pairs continued working nominally after the heater
element failure. The voltages observed by the thermocouples are only weaker and contain more noise, but they have the same dynamic as during sols 1-45, when the heater element was intact. Sec. 3.5 presents a method for calibrating the quadrant sensor voltages with the reconstructed wind directions from sols 46-377.

After calibrating the voltages of the quadrant sensor, the wind directions can be solved for sol 377 and beyond. With the reconstructed wind direction and one stream of intact wind component data the wind speeds can be solved for sol 377 and
onwards. The complete VL1 mission was reconstructed using the calibrated quadrant sensor signals, which were calibrated using the Nusselt number wind direction estimates from segment one.

## 3.2  Calibration of the wind sensors

The wind direction can be obtained from the Nusselt numbers. This is because the Nusselt numbers are dependent of the wind components normal to the hot-film sensors. The first segment of the wind reconstruction algorithm relies on the assumption
that the wind sensors get similar values for Nusselt numbers during the period of interest as during the sols 1-45. The values of Nusselt numbers for both wind sensors were examined from arbitrary sols from the sol interval 46-377 of the VL1 mission. The results from this spot check were encouraging as the Nusselt numbers of the wind sensors remained the same order of magnitude during the examined sols. The method of using the Nusselt numbers for estimating the wind direction was first studied by (Murphy et al., 1990). The method used here is similar to method used in (Murphy et al., 1990), but it contains less
subjectivity in the process as the wind directions are not reconstructed mechanically by hand.

The reconstruction algorithm begins by computing first the calibration functions $F_i$, for different wind velocity classes, from the Nusselt numbers $Nu_1$ and $Nu_2$ measured by the wind sensors. The calibration functions $F_i$ are defined the same way as in (Murphy et al., 1990):

$$F = \frac{(Nu_1 - Nu_2)}{\sqrt{Nu_1 Nu_2}}. \tag{3}$$

After calculating the values for $F$ by Eq. (3), they were binned into one degree sized bins. Then, a 12th order polynomials were fitted to the binned values calculated with Eq. (3). The equation for the calibration function $F_i$ was obtained from the fitting process. Various values for the order $n$ of fitted polynomial were examined and the most functional value was determined to be $n = 12$. For this value of $n$, the mean absolute difference of the reconstructed angle, and the SANMET-provided "correct" angle from sols 1-45 was the smallest.
Because the Nusselt numbers depend on the wind velocity, the velocities were divided into four different velocity classes. For each velocity class the calibration function was obtained using the method described earlier. Tab. 1 presents the different velocity classes for which the calibration functions $F_i$ were determined. The wind velocities, which were greater than $50\,\mathrm{ms}^{-1}$,


**Table 1.** The velocity classes of the wind reconstruction algorithm.

| Class number | Wind velocity |
|:---:|:---:|
| 1 | $< 2 \ \mathrm{ms}^{-1}$ |
| 2 | $2 - 4 \ \mathrm{ms}^{-1}$ |
| 3 | $4 - 8 \ \mathrm{ms}^{-1}$ |
| 4 | $8 - 50 \ \mathrm{ms}^{-1}$ |

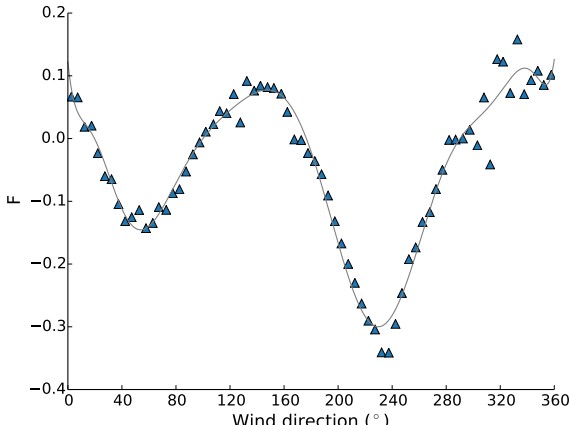

**Figure 6.** Calibration function $F_1$ for velocity class 1.

were quite rare during the first 45 sols of VL1. Thus, there were not enough data of the Nusselt numbers from these velocities to make good statistics, and the velocities were ignored.

The calibration functions $F_i$ for each wind class are plotted in Figs. 6, 7, 8, 9 with gray colored curves. Although there exists slight overfitting effect in the calibration functions in Figs. 6, 8, 7, the calibration functions can distinguish the wind directions from different F values as the data is distributed uniformly to the full angle interval. For the calibration in Fig. 9 the data is not distributed evenly to the complete angle interval, which reduces the accuracy of the wind direction estimates.

The data for the calibration functions in Figs. 6, 8, 7, 9 were obtained from the first 45 sols, when the quadrant sensor was still fully functional, and it consisted of 53,229 measurement samples. Most of the samples were from conditions where the wind velocity was less than $8.0 \ \mathrm{ms}^{-1}$.

To reconstruct the wind directions of a complete sol, the algorithm calculates for each measurement sample the value of function $F$ using Eq. (3) and determines the velocity class of the sample. With the calculated $F$ value, the roots $\theta_i$ of 12th order polynomial were calculated by solving

$$\sum_{i=0}^{12} a_i \theta^i - F_{\mathrm{sample}} = 0 \qquad (4)$$





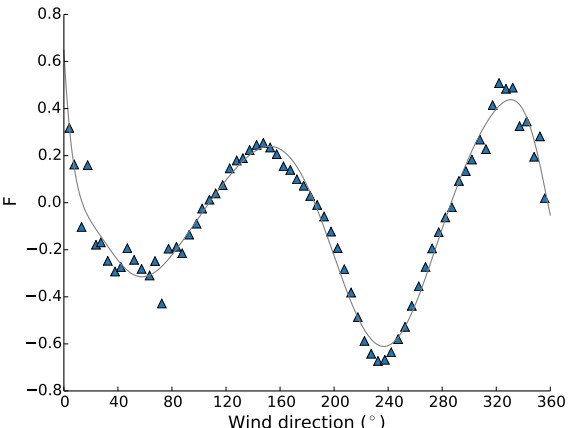

**Figure 7.** Calibration function $F_3$ for velocity class 3.

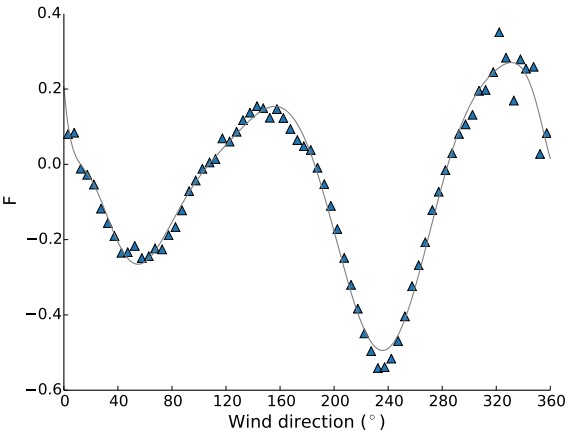

**Figure 8.** Calibration function $F_2$ for velocity class 2.

The coefficients $a_i$ of the polynomial were obtained from the calibration function fit. The roots $\theta_i$ of Eq. (4) were used as candidate angles for wind direction. This set of angles usually contains either two-fold or four-fold ambiguity, which is resolved using either the quadrant sensor signals, or time continuity. It is possible that for a certain sample there are no roots for Eq. (4) of the calibration functions, therefore the candidate angles for wind direction can not be solved. In this situation an error value is placed as the value of the particular sample.

If the absolute value of the bias corrected quadrant sensor voltages exceeded the threshold value (0.05 mV), the correct quadrant for wind direction was established from the quadrant sensor signals using a look-up table presented in Tab. 2. The look-up table is based on the intended operating principle of the quadrant sensor and the orientation of the VL1 wind measure-




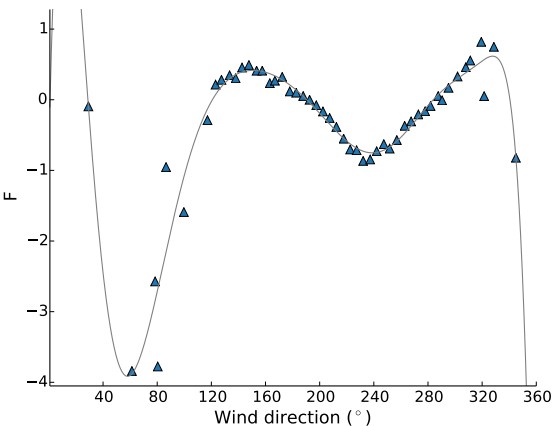

**Figure 9.** Calibration function $F_4$ for velocity class 4.

**Table 2.** VL1's look-up table of the developed wind reconstruction algorithm.

| Angle range | $V_{QS_1}$ | $V_{QS_2}$ |
|---|---|---|
| $350° - 80°$ | $< 0$ | $> 0$ |
| $80° - 170°$ | $< 0$ | $< 0$ |
| $170° - 260°$ | $> 0$ | $< 0$ |
| $260° - 350°$ | $> 0$ | $> 0$ |

ment instruments, which is shown in Fig. 10. The quadrant sensor thermocouple pair $QS_1$ has positive voltage values, when the wind is from TC-1 to TC-2. Respectively the thermocouple pair $QS_2$ has positive voltage values, when the wind is from TC-3 to TC-4.

The ambiguity of the wind direction is settled by selecting from the set of candidate angles the specific angle that is in the

5    correct wind quadrant, as determined by the quadrant sensor voltages. If there are many candidate angles in the same wind quadrant, the wind direction is arbitrarily selected to be the smallest angle from the set of candidate angles.

If the quadrant sensor voltages did not exceed the threshold value, time continuity is used for determining the wind direction. Time continuity works as follows: the chosen candidate angle is the one nearest to the last determined angle, but only if the time difference between that one and the current one is less than one hour. This principle is sufficient when the elapsed time

10    between the measurement samples is not too long, as wind often exhibits continuous behavior. However, if the time between the two samples exceeds the threshold of one hour, the use of time continuity is likely not valid and in these cases the Dynamic Wind Table (DWT) is used to determine the correct angle. The operation principle of the DWT is described in Sec. 3.3. The operation of the first segment in the wind reconstruction algorithm is summarized in Alg. 1.




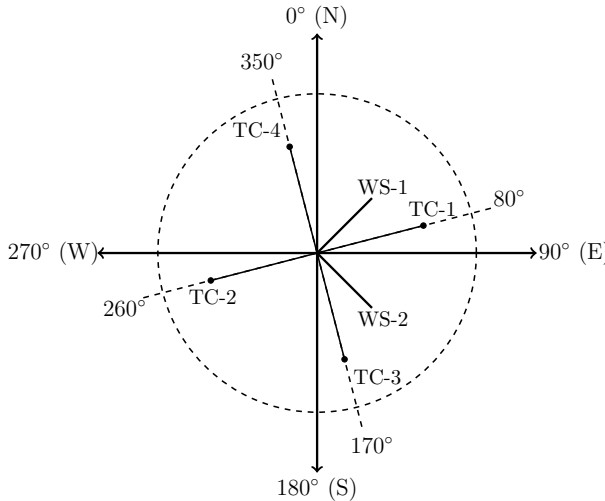

**Figure 10.** The orientation of VL1's wind sensor assembly, adapted from (Murphy et al., 1990). The degree coordinate axis refers to the Martian planetary directions. The abbreviations TC and WS stand for thermocouple and wind sensor respectively.

### 3.3 Dynamic Wind Table

Time continuity was used for approximately 60% of the reconstructed angles of VL1 during sols 46-377. DWT was developed to determine the wind direction when the use of time continuity was not possible. Notably, DWT is capable of taking into account the Mars' seasonal variations in wind direction.

5    DWT's was implemented using a hash table, with LLT in hours as the key. The value of the table is a queue containing the mean value of the reconstructed angles for the corresponding hour. When reconstructing a new sol, new values for hourly mean wind direction were calculated and then added to the DWT's queue. DWT's queue for hourly mean wind direction will hold data at most from the last 10 reconstructed sols. The structure of the DWT is presented in Fig. 11.

When the use of time continuity was required due to the quadrant sensor voltages being too low, the time difference in hours

10  between current sample and last measured sample was solved. If the time difference of samples is more than one hour, the time value was used to obtain the mean wind direction of the hour from those of the last 10 sols where data from the hour in question was recorded. A maximum allowed rate of wind direction change was defined to prevent error values filling the DWT. If the allowed rate of change is exceeded, the wind direction is not added to the DWT, thus preventing outliers from distorting the "correct" average values.

### 15  3.4 Bias correction of the quadrant sensor voltages

The bias corrections of the thermocouple pairs QS1 and QS2 do not stay constant during the VL1 mission. There exists a drift in both of the pairs' voltages. Therefore a diurnal bias correction for QS1 and QS2 is required to distinguish more reliably the

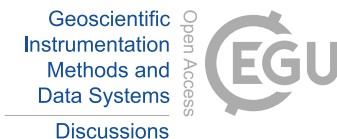

---

**Algorithm 1** The algorithm for wind reconstruction in the first segment (sols 46-377).

---

**Input:** Wind measurement data and the pre-calculated calibration functions $F_i$

**Output:** The reconstructed wind directions for each measurement sample

**for all** data samples **do**

    Calculate the $F$ value of the sample

    Determine the velocity class of current sample

    Solve the roots of calibration function $F_i$ for the correct velocity class

    **if** $F_i$ has roots **then**

        **if** $|V_{\mathrm{QS}_1} - \mathrm{bias}_{\mathrm{QS}_1}| > 0.05$ mV **and** $|V_{\mathrm{QS}_2} - \mathrm{bias}_{\mathrm{QS}_2}| > 0.05$ mV **then**

            Identify the wind quadrant using the look-up table

            Select angle in the correct quadrant from the set of candidate angles

            $t_{\text{last sample}} = t_{\text{current sample}}$

        **else**

            $\Delta t = t_{\text{current sample}} - t_{\text{last sample}}$

            **if** $\Delta t > 1$ h **then**

                Use time continuity and select the angle nearest to the last determined angle

            **else**

                Select the angle nearest to the angle given by the DWT for current hour

            **end if**

        **end if**

    **else**

        Set error value for this sample

    **end if**

**end for**

---

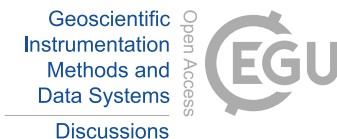

Figure 11. The DWT schema.





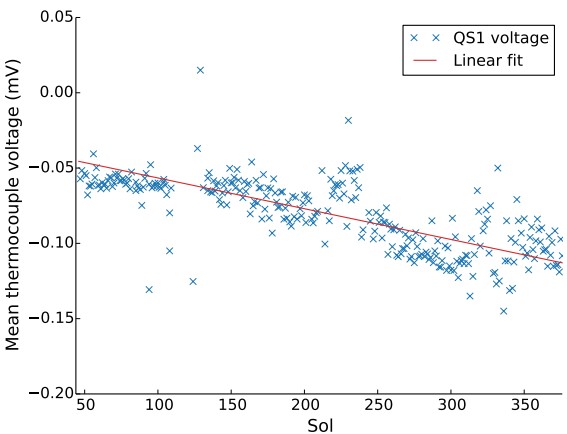

**Figure 12.** Bias correction of QS1.

correct quadrant sensor signals from noise. The samples used in determining the bias corrections were from nighttime records during light wind conditions.

The criterion for nighttime was defined to be that the sample is measured between 00:00-06:00 LLT, and the criterion for light wind conditions was that the wind velocity is less than $0.8~\mathrm{ms}^{-1}$. The voltage values of the samples, which meet these two criteria, were filtered from all the measured samples, and the diurnal mean values of voltages were calculated. A linear fit was made to the calculated mean voltages to obtain the diurnal bias correction for those sols where the wind direction is reconstructed. The diurnal mean voltage values of the thermocouple pairs and linear fit are presented in Figs. 12 and 13. Even though the thermocouple pairs QS1 and QS2 are almost identical, the bias voltage of QS1 changed more during sols 46-377 than QS2 bias voltage.

## 3.5 Calibration of the quadrant sensor

In the second segment of the wind reconstruction algorithm, the recalibration of the quadrant sensor signals is required. For calibrating the voltages of the quadrant sensor, a second calibration function, called $R$, was defined as

$$R = \frac{V_{\mathrm{QS}_1}}{V_{\mathrm{QS}_2}}, \tag{5}$$

which is the ratio of the thermocouple voltages $V_{\mathrm{QS}_1}$ and $V_{\mathrm{QS}_2}$. When the calibration function $R$ is plotted as a function of wind direction, a tangential function is observed, as shown in Fig. 14. The tangential function arises from the geometry of the quadrant sensor. There exists angles $180°$ apart for both thermocouples where the wind velocity vector is perpendicular to the thermocouple pair QS2. In this case, the temperature difference, measured by the thermocouple, is zero, and the thermocouple





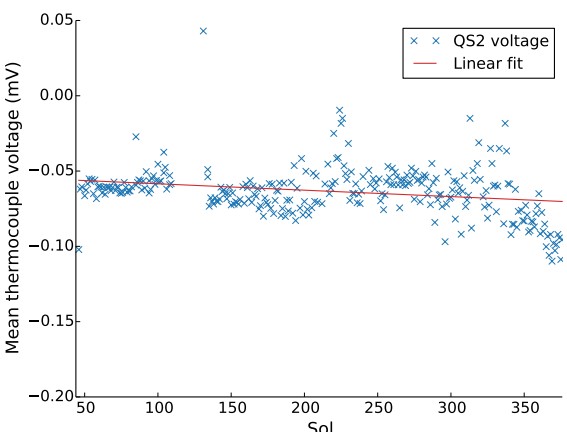

**Figure 13.** Bias correction of QS2.

can not detect the wind flow from this direction. For fitting a tangential function into the $R$-data of the quadrant sensor the following model was used

$$R(\theta) = A\tan(B + \theta) + C \qquad (6)$$

The fitting of Eq. (6) is a nonlinear optimization problem, for which the least-squares method (LSQ) was used. The task was to minimize the squared difference between data and the model values. The results of fitting Eq. (6) into the quadrant sensor data is shown in Figs. 14 and 15. The fit in Fig. 15 is not as good as in Fig. 14, because in the data of Fig. 15 there exists artifacts caused by yet unknown reasons. In general the shape of the data in Fig. 15 seems similar to the data in Fig. 15. The values obtained for parameters $A$, $B$ and $C$ during sol intervals 1-45 and 46-376 are presented in Tab. 3.

**Table 3.** The values of parameters $A$, $B$ and $C$ obtained from the LSQ-fits.

| Sol interval | $A$ | $B$ | $C$ |
|---|---|---|---|
| 1-45 | -0.8405 | 0.1814 | 0.0545 |
| 46-376 | -0.5429 | 0.1675 | -0.1017 |

After obtaining all the values for Eq. (6) parameters, the fitted tangential model is used for predicting the wind directions for new values measured for the ratio of voltages $R$ by the quadrant sensor. The calibration function defined in Eq. (6) can be used to reconstruct the wind directions during the complete VL1 mission, because only the heater element in the quadrant sensor failed. The thermocouples remained operating nominally; therefore, the ratio $R$ of quadrant sensor voltages should obtain values in the same order of magnitude during the whole VL1 mission. The distribution of values, obtained for $R$, is





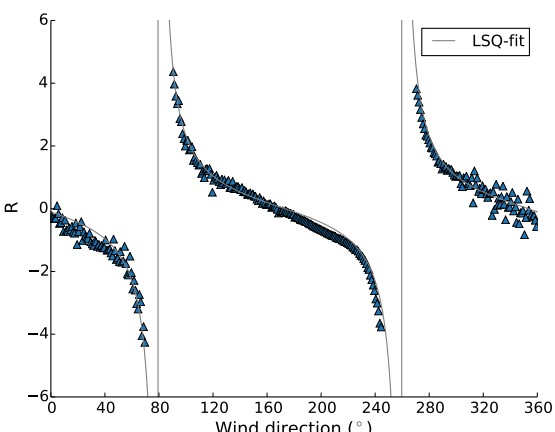

**Figure 14.** Calibration function $R$ during sols 1-45.

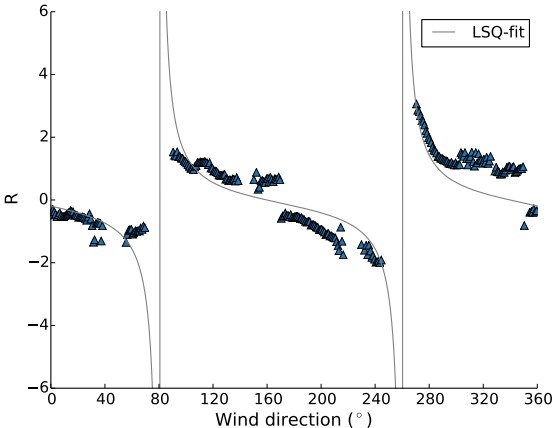

**Figure 15.** Calibration function $R$ during sols 46-377.

shown in Fig. 17. The algorithm for wind reconstruction in segment 2 is similar to the algorithm presented in Alg. 1, except that the $R$-value is calculated using Eq. (5) instead of $F$-value.

## 4 Validation of the wind reconstruction algorithm

### 4.1 Error analysis of VL1 sols 1-45

5 The error analysis of the wind reconstruction algorithm is done by comparing the data produced by the the algorithm with the VL1 SANMET wind data. Because the VL1 wind measurement instruments remained fully functional for the first 45 sols





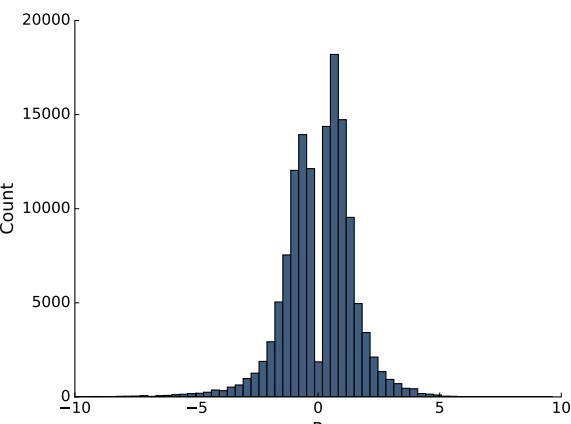

**Figure 16.** The distribution of $R$ values during sols 46-377.

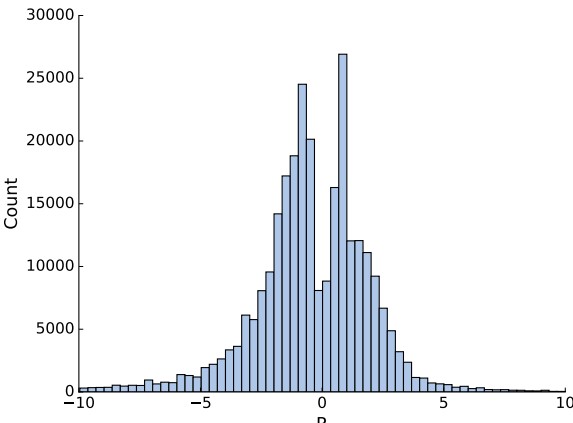

**Figure 17.** The distribution of $R$ values during sols 378-2245.

of VL1 mission, it is assumed that the data for that period is correct and can be used as a reference for the comparison. The behaviour of the algorithm was tracked for every reconstructed sol, and data for the error analysis were gathered simultaneously with the reconstruction. The key indicator for the performance of the algorithm is the absolute difference between SANMET and reconstructed angles

$$5 \quad \Delta\theta = \left|\theta_{\text{SANMET}} - \theta_{\text{Algo}}\right|. \tag{7}$$

The angle difference of Eq. (7) were calculated for every measurement sample of the reconstructed sols, and for each sol reconstructed, a mean value for the angle difference and standard deviation were calculated. The result of these calculations





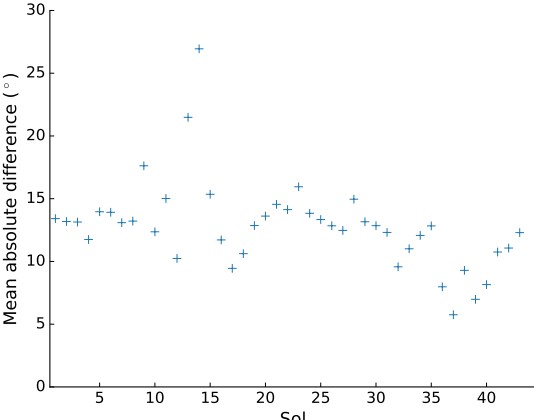

**Figure 18.** The diurnal mean angle difference.

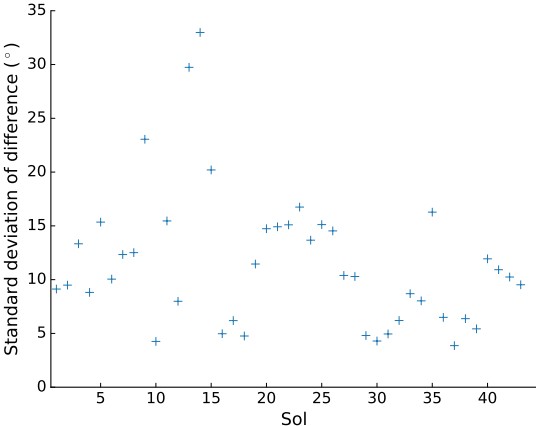

**Figure 19.** The diurnal standard deviation of the angle difference.

for the VL1 mission are shown in Fig. 18. The mean value of the mean of the angle differences is $12.8°$ and the mean value of the standard deviations is $11.5°$. By visually inspecting Fig. 18 the means of the angle difference vary mostly in the range from $5°$ to $15°$, which is quite reasonable variation for determining the correct point of compass for the Martian tides.

The reconstruction of wind direction is very accurate during the first 45 sols of VL1 mission. The reconstructed wind speed shows reasonable behaviour as the wind speed begin to increase after the sunrise around the seventh hour of LLT. The standard deviations in the reconstructed wind speed are much larger compared to the SANMET wind speeds. In general, the hourly mean of the reconstructed wind speed is greater than the SANMET determined mean value. However, the SANMET determined mean value for wind speed sets in the error limit of the reconstructed wind speed in all hours, except 20 and 21. In





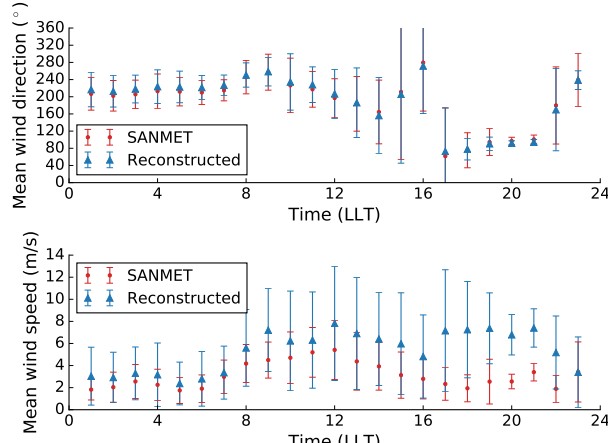

**Figure 20.** The hourly mean wind direction and speed from SANMET and reconstruction during sols 1-45.

Fig. 20 the hourly means of SANMET determined wind directions and speeds and the reconstructed wind directions and speeds are shown during the first 45 sols of VL1 mission. The errorbars in Fig. 20 are the standard deviations of the measurements done during the specific hour. In addition to the hourly mean wind direction and speed, a point-by-point comparison between the SANMET and reconstructed data are shown in Figs. 21 and 22. Fig. 21 presents the comparison between wind direction and

5 speed of SANMET and reconstruction from the forenoon of sol 3. The mean absolute difference between the SANMET and reconstructed wind direction was $9.1°$ during the 6 minute time period. The reconstructed wind speeds have a high-speed bias caused by the wind direction being almost parallel to the WS1. Fig. 22 displays SANMET and reconstructed wind directions and speeds during a one hour period of the midday of sol 41. During the one hour period, the mean absolute difference between SANMET and reconstructed wind directions were $12.2°$.

The method, how a particular measurement sample was determined, was also recorded during the reconstruction process. In total of 45 VL1 sols were reconstructed for the analysis. During these sols 53,070 samples were measured. During the reconstruction process 52,570 samples ($99.1\%$) were determined using the look-up table in Tab 2, 500 samples ($0.9\%$) were obtained using time continuity. The look-up table was used for the majority of the time to determine the wind direction as there was no threshold value for voltage signals, because the signals were bias-corrected and assumed therefore to be correct. The

situations, when time continuity was applied in the reconstruction process, occurred during nighttime hours (00:00-06:00), when the Sun's radiation was not strong enough to heat the quadrant sensor. In Figs. 23, 24, the histograms of the mean difference of reconstructed and SANMET angles as well as the standard deviation of the angle difference are presented.

## 4.2 Slope winds of VL1 area

Another validation method for the reconstruction algorithm is based on the physical fact that there should exist slope winds

in the reconstructed wind data. The VL1 landed on a slope rising to south and west. In this case the slope winds in VL1





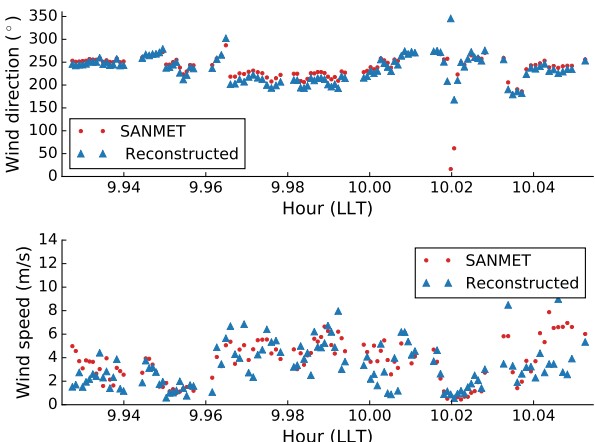

**Figure 21.** Wind direction and speed during 6 minutes time period from the forenoon of sol 3.

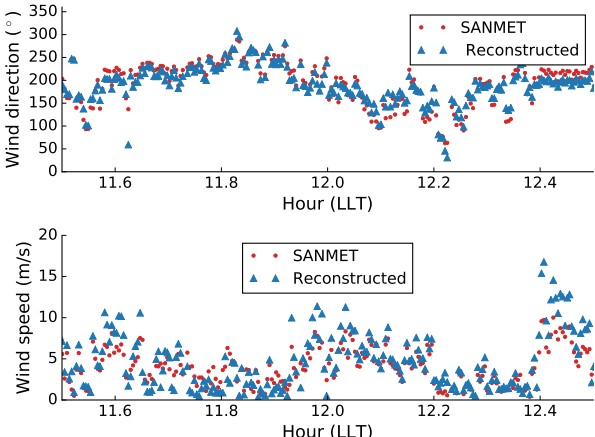

**Figure 22.** Wind direction and speed during one hour at the midday of sol 41.

area will form when the nocturnally cooled dense air is accelerated down the slope by gravity. Therefore, in nocturnal hours, the direction of wind should be in the interval $180°$-$270°$, in which case the wind is from top part of the slope. Soon after the sunrise, the direction of wind should change, so that the daytime winds are anabatic. The anabatic winds are accelerated upslope by sun-heated warm slopes. The daytime wind directions should regularly be in the interval $0°$-$90°$. The slope winds

5    are typically observed in Martian summer as the conditions for observing the slope winds are optimal. The conditions for observing the slope wind are a terrain sloping over a large area, strong diurnal temperature variations and weak ambient winds which are typical on Mars in summer (Savijärvi and Siili, 1993).





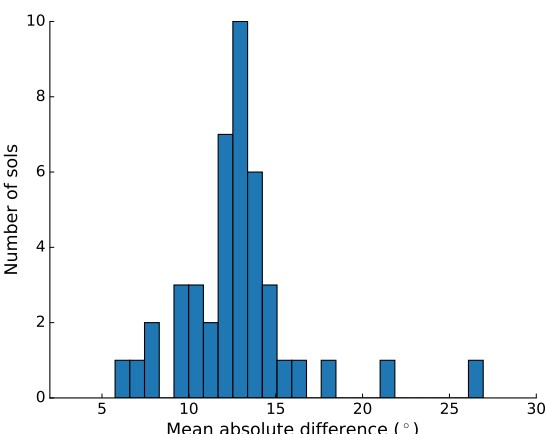

**Figure 23.** Mean absolute difference of the angles during sols 1-45.

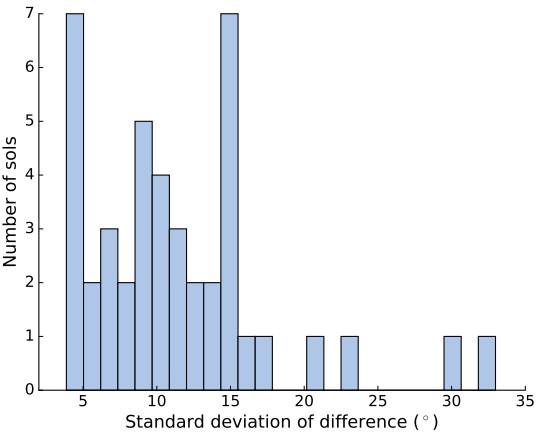

**Figure 24.** Standard deviation of the angle difference during sols 1-45.

The statistical distribution of wind directions were examined during the sols 1-45, the distributions are presented in Fig. 25 with ten degree bin sizes. The bin with the largest count for SANMET wind directions is 200°-210° bin and for the reconstructed wind directions the bin is 220°-230°. Taking into account that the mean difference between SANMET wind directions and reconstructed directions was 12.8° during sols 1-45, the locations of the peak maximums are consistent with each other. Inspecting the left side tail of both distributions, there exists a small local maximum approximately between 80°-100°. The small local maximum is likely caused by the daytime anabatic winds, which flow upslope.




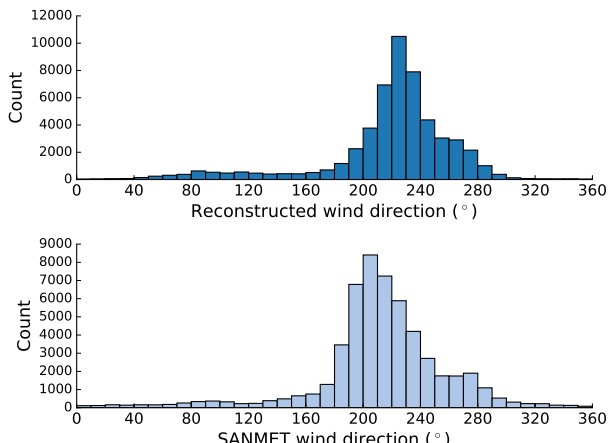

**Figure 25.** Statistical distributions of wind directions during VL1 sols 1-45.

## 5  Reconstructed sols of VL1

The algorithm was used to reconstruct all available VL1 sols, and the results from the reconstruction are presented in this section. The statistical distribution of wind directions is shown in a histogram in Fig. 26, for studying the existence of slope winds in the VL1 area. In the figure the wind directions are divided into 10 degree sized bins. The distribution of the wind

directions determined by SANMET is shown in Fig. 26 as a comparison with the reconstructed wind directions. The SANMET method for determining the wind direction is not reliable, because it does not take into account the decay of the quadrant sensor heater element and likewise the failure of one of the wind sensors.

Two clear peaks are visible in Fig. 26 at the reconstructed wind directions. One peak is located between 80°-120° and the other is in interval 240°-300°. The peak, which is in the angle range 80°-120° is much sharper compared to the peak in range

240°-300°. The peak in range 240°-300° corresponds to the nocturnal wind, which is directed downslope and the peak in range 80°-120° is the daytime anabatic upslope wind.

Two arbitrarily selected sols from the VL1 mission are presented to illustrate the reconstructed wind measurements. The sols selected are 74 and 1408. The quadrant sensor of VL1 had failed before the sol 74 in Fig. 27, but both of the wind sensors were still intact. During the sol 1408, both the quadrant sensor heater and one of the hot-film wind sensors had already failed.

Thus the SANMET wind speed in Fig. 28 contains only erroneous values.

The hourly mean of wind directions and speeds for both sols are shown in Figs. 27 and 28. Both of the inspected sols occur during Martian late summer, sol 74 being the year one summer and the sol 1408 the third year summer of VL1 mission. Thus the conditions for observing slope winds should be favourable. During the nocturnal hours the wind direction is in range 260°-280°, which roughly corresponds to the wind flowing from the upslope. After the sunrise the wind rotates about 180°

and begins to flow from the lower part of the slope.





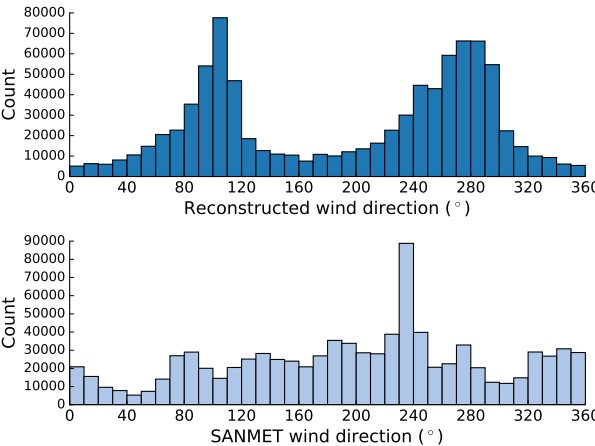

**Figure 26.** The comparison between SANMET distributions of wind directions and reconstructed during the complete VL1 mission.

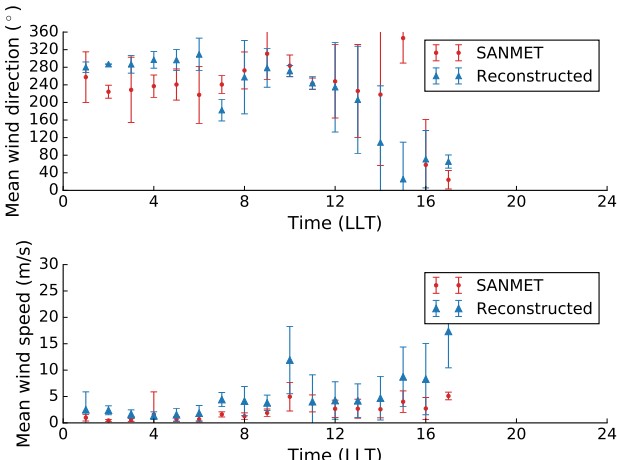

**Figure 27.** Sol 74.

## 6 Summary and discussion

The article focused on developing an algorithm to reconstruct the wind measurements during the complete VL1 mission. VL1 performed wind direction and speed measurements on the surface of Mars for 2245 sols, thus the dataset produced by the wind reconstruction is significant in its size.

5     The wind measurement system of VL1 consisted of two orthogonal hot-film wind sensors and a quadrant sensor for solving the ambiguity in wind direction. The quadrant sensor failed during sol 45 and one of the wind sensors broke down during sol 378, thus only one of these three instruments remained fully intact for the VL1 mission. However, with the algorithm described in this report, it was possible to reconstruct the wind measurements with a reasonable accuracy.





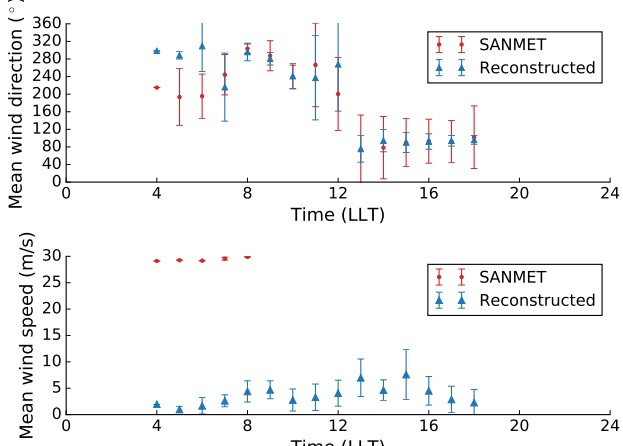

**Figure 28.** Sol 1408.

The wind reconstruction was completed in two segments. In the first segment the quadrant sensor signals from sols 1-45 were used to calibrate Nusselt numbers of the hot-film wind sensors. The wind sensors were then used to estimate wind directions during sols 46-377. With the estimated wind directions the quadrant sensor was recalibrated and was then used to reconstruct all the wind directions from VL1 mission. The reconstruction of wind speed was also possible with one wind velocity component and the wind direction. The results from the reconstruction of the wind speeds are not always very reliable, because the reconstruction of wind speed required both knowledge of the wind direction, and the wind velocity component from the nominally working wind sensor. The wind directions contain variable amounts of error between different sols. Therefore, the reconstruction quality of the wind speed is weaker for sols with more error in the wind direction.

The developed algorithm for wind reconstruction shows the presence of slope winds in the VL1 area. The precision of the algorithm compared to the data measured by fully functional VL1 wind measurement instruments is reasonable. On average the mean of the absolute difference between wind direction was determined to be $12.8°$.

The new wind reconstruction algorithm developed in this article extends the amount of available sols of VL1 from 350 to 2245 sols. The reconstruction of wind measurement data enables the study of both short term phenomena, such as daily variations in wind conditions or dust devils, as well as the study of long term phenomena, such as the seasonal variations in Martian tides.

*Acknowledgements.* The authors are are thankful for the Finnish Academy grant #131723.



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
