# Peer review of "Wind Reconstruction Algorithm for Viking Lander 1"

_Geoscientific Instrumentation, Methods and Data Systems, 2016_

## Referee Comment (RC1) · Anonymous Referee #1 · 27 Mar 2017

**1. General Comments**

The authors present a new algorithm to reconstruct Viking Lander 1 (VL1) wind measurements from sol 350 to 2245 (last sol with measurements). Measurements during this period are currently missing from NASA's Planetary Data System (PDS) due to failures in the quadrant sensor and in one of the two hot-film wind sensors. Reconstructed VL1 wind measurements are extremely relevant to the field due to the scarcity of measurements of this kind and their importance to shed light on atmospheric phenomena ranging from turbulent to climatic time-scales. The methodology developed to reconstruct wind measurements is well presented and robust, and its performance reasonable. This reviewer's recommendation is that the paper may be accepted for publication after a few minor issues are addressed (see specific comments) and the use of English language is improved.

2. Specific Comments

Page 1, L3: What is the duration of the initial phase of the mission?

Page 1, L3, L10, L11 and L21: Was sol 45th the last sol of the initial phase? If one of the wind sensors broke down during sol 378, why PDS-available wind data extend only to sol 350?

Page 1, L21: 'Extended' from NASA's PDS?

Page 1, L29: I suggest writing '(Chamberlain et al., 1976)' after 'meteorological experiments' and '(Soffen et al., 1977)' after 'payload'.

Page 1, L33: I suggest deleting the reference to Soffen (1977) from this line.

Page 1, 3rd Paragraph: Please, rewrite or clarify this sentence. I suggest 'The VL1 data set enables the study of various meteorological phenomena ranging from diurnal to seasonal time scales using measurements of temperature, pressure and wind direction'.

Page 1, 5th Paragraph, first sentence: I suggest to replace 'poor' by 'limited'.

Page 1, 5th Paragraph: I suggest mentioning that the MPF wind data is not available in the NASA PDS.

Page 1, 5th Paragraph: Phoenix's "telltale" measurements should be mentioned [Holstein-Rathlou et al., 2010].

Page 1, 5th Paragraph: I suggest mentioning that the MSL wind sensor can still be used to retrieve wind speeds and accurate wind directions when wind comes from the hemisphere in front of the rover Newman et al., (2017).

Page 2, 6th Paragraph: I suggest adding a summary of Section 2 and 4 (instead of just 4.1).

Page, 3, L37: Which algorithm, yours? If so, I suggest deleting these two sentences

from this subsection.

Page 4, L55-57: I suggest rewriting this sentence.

Page 6, L26: Have the authors quantified the impact of a threshold different from one hour on the results?

Page 6, 40: LLT is not defined.

Page 7, Eq. (5): VQS1 and QS1 are used interchangeably across the manuscript. I suggest to stick to one of them.

Page 8: Fig. 16 is not referenced.

Page 9: Fig. 19 is not referenced in the text.

Page 10, Figs. 21 and 22: Why in particular are these sols&LLT chosen? What exactly makes both figures representative of the performed reconstruction?

Page 11, L1-15: The discussion made in these lines seems to remain open as it is not supported by any Figure included in this subsection.

Page 11, L56: Provide solar longitude values (Ls) values for both sols.

3. Technical Corrections

Page 8, L15: Replace 'Fig. 15' by 'Fig. 14'.

---

## Author Comment (AC1) · 20 Apr 2017

Specific comments:

1. *"Page 1, L3: What is the duration of the initial phase of the mission?"*

The "initial phase of the mission" is corrected to "first 45 sols of the mission".

2. *Page 1, L3, L10, L11 and L21: Was sol 45th the last sol of the initial phase? If one of the wind sensors broke down during sol 378, why PDS-available wind data extend only to sol 350?*

The PDS-available data is produced with the algorithm by (Murphy et al., 1990). There was no mention in (Murphy et al., 1990), why the reconstruction was not extended to sol 378.

[Figure]

3. *Page 1, L21: 'Extended' from NASA's PDS?*

The tools extend the data available in NASA's PDS. The sentence was restructured to make it more clear.

4. *Page 1, L29: I suggest writing '(Chamberlain et al., 1976)' after 'meteorological experiments' and '(Soffen et al., 1977)' after 'payload'.*

The citations were changed as the reviewer suggested.

5. *Page 1, L33: I suggest deleting the reference to Soffen (1977) from this line.*

The reference was removed from this line.

6. *Page 1, 3rd Paragraph: Please, rewrite or clarify this sentence. I suggest 'The VL1 data set enables the study of various meteorological phenomena ranging from diurnal to seasonal time scales using measurements of temperature, pressure and wind direction'.*

The sentence was rewritten as the reviewer suggested.

7. *Page 1, 5th Paragraph, first sentence: I suggest to replace 'poor' by 'limited'.*

The word 'poor' was changed to 'limited'.

8. *Page 1, 5th Paragraph: I suggest mentioning that the MPF wind data is not available in the NASA's PDS.*

Mention that the MPF wind data is not available in the NASA's PDS was added.

9. *Page 1, 5th Paragraph: Phoenix's "telltale" measurements should be mentioned [Holstein-Rathlou et al., 2010].*

A mention of Phoenix's "telltale" measurements was added.

10. *Page 1, 5th Paragraph: I suggest mentioning that the MSL wind sensor can still be used to retrieve wind speeds and accurate wind directions when wind comes from the*

*hemisphere in front of the rover Newman et al., (2017).*

A mention was added that the MSL wind sensor can be used, when the wind comes from the hemisphere in front of the rover.

11. *Page 2, 6th Paragraph: I suggest adding a summary of Section 2 and 4 (instead of just 4.1).*

Summary of sections 2 and 4 were added.

12. *Page, 3, L37: Which algorithm, yours? If so, I suggest deleting these two sentences from this subsection.*

The two sentences were deleted from the subsection.

13. *Page 4, L55-57: I suggest rewriting this sentence.*

The sentence was rewritten.

14. *Page 6, L26: Have the authors quantified the impact of a threshold different from one hour on the results?*

Different threshold values were examined and the threshold value of one hour yielded the minimum mean difference between the wind directions of SANMET and the reconstruction algorithm during sols 1-45.

15. *Page 6, 40: LLT is not defined.*

The Lander Local Time (LLT) is now defined, where it was first used.

16. *Page 7, Eq. (5): VQS1 and QS1 are used interchangeably across the manuscript. I suggest to stick to one of them.*

The term $QS1$ refers to the thermocouple pair 1 and $V_{QS1}$ refers to the voltage value of the thermocouple pair 1. We felt that keeping these separate is important for clarity.

17. *Page 8: Fig. 16 is not referenced.*

Reference to Fig. 16 was added.

18. *Page 9: Fig. 19 is not referenced in the text.*

Reference to Fig. 19 was added.

19. *Page 10, Figs. 21 and 22: Why in particular are these sols & LLT chosen? What exactly makes both figures representative of the performed reconstruction?*

The figures were chosen arbitrarily from the part of the VL1 mission, where the wind sensors were fully functional.

20. *Page 11, L1-15: The discussion made in these lines seems to remain open as it is not supported by any Figure included in this subsection.*

We have expanded the discussion in 4.2.

21. *Page 11, L56: Provide solar longitude values (Ls) values for both sols.*

The solar longitude values ($L_s$) were added to Figs. 20, 21 and 22.

Technical corrections:

1. *Page 8, L15: Replace 'Fig. 15' by 'Fig. 14'.*

Reference to the correct figure was added.

Thank you very much for your efforts in reviewing this article.

**T. Kynkäänniemi et al.**

**Interactive comment on Geosci. Instrum. Method. Data Syst. Discuss., doi:10.5194/gi-2016-41, 2017.**

---

## Referee Comment (RC2) · J. Murphy (Referee) · 24 Apr 2017

Derivation of a wind vector data set from the Viking Lander 1 wind sensor system is a tremendous addition to martian surface meteorology studies. The Viking Lander 1 meteorology data set remains the most time extensive surface-obtained martian atmospheric data set, but the lack of the wind measurements to accompany the pressure and temperature and opacity data sets has been a substantial deficiency.

The manuscript describes the methodology developed and applied to employ degraded wind directional sensor signals and additionally wind speed sensor signals, and presents some comparison of the derived winds using this/these methodologies with wind speed and direction measurement from when the full sensor system was operable. Additionally, derived winds are assessed in the context of slope-induced winds

anticipated at the VL1 location.

Specific Comments: There are previously published VL1 winds derived following the quadrant sensor failure (Murphy et al., 1990). The manuscript does not provide any comparisons between those previous derived wind vectors and those derived using the method in this manuscript; such comparisons are warranted, as is some additional measurement-by-measurement comparison with the SANMET derived winds during the initial 45 sols (in addition to hourly averaged comparison, Figures 25 & 26). The impact of the manuscript will be strengthened with the inclusion of presentation of some derived point-by-point wind speed and direction, in addition to the hourly averaged presentation included in the current draft. Will the derived wind data set be made publicly available in its entirety?

There is an apparent high-wind-speed bias that arises from the manuscript's methodology when compared to SANMET winds during sols 1-45. The manuscript does address this result but more attention is warranted to provide a more substantial basis for believing the derived wind speeds and directions.

It would be helpful if Figure 3 included Sol 45 quadrant sensor signals to illustrate what a fully uncompromised sol's measurements exhibit. The presentation would benefit from displaying the nominal quadrant sensor signal during a complete sol which subsequently transitioned to the instrument behavior change evident during the first few hours of Sol 46 and Sol 47 which then became persistent during Sol 48.

The substantial comparison within the paper of the newly derived wind vector results with wind vectors derived applying the SANMET software to the degraded sensor signals is unwarranted. [Figures 24, 25, and 26.] The SANMET software was designed to operate with signals from fully functional instruments. There is no doubt that SANMET-derived wind speeds and directions from the compromised instruments will be flawed, and using such flawed results as a comparison with the newly derived wind speeds and directions does not itself provide validation of the newly derived results. Rather, it

is better to compare the newly derived winds with anticipated environmental conditions and their presumed physical driving of the winds that were experienced.

For instance, the winds experienced at VL1 during the two global scale dust storms that occurred during the first year of the mission, initiating at Ls ~205 and ~270 (sols ~210 and ~315), are theoretically expected to have exhibited a semi-diurnal rotation of the wind vector, which the derived winds from Murphy et al (1990) were in agreement with.

The hodograph figures in Murphy et al illustrate this wind vector rotation arising from amplified thermal tides. Also, Viking Lander 1 camera images of the landing site provided evidence of surface material motion believed to be due to wind stress, requiring fast wind speeds from a direction indicated by the direction of material motion (Sagan et al, JGR, 82, September 1977, 4430-4438; Moore, H., JGR, 90, November 1985, 163-174).

In Section 3.1 the word 'segment' is invoked to describe each of the two time intervals (sols 46-377, sols 378-2245) during which the two specific wind derivation methodologies are implemented/applied. Since 'segment' is frequently used to describe a portion of a physical structure rather than a time interval, I suggest considering replacing 'segment' with a word that unambiguously indicates time, such as 'stage'. Stage 1 could correspond to the failed quadrant sensor during sols 46-377 while the wind sensor continued operating, while Stage 2 could correspond to the subsequent failed Wind Sensor element condition.

Since the concept of Nusselt number (introduced on Page 3) is very important to the paper's discussion of the wind sensor signals, I recommend providing a definition of Nusselt Number within the text.

In Section 2.4, final sentence of the 2nd paragraph, the statement '.. a significantly higher temperature than the ambient temperature' occurs, but there is no declaration as to the necessary magnitude of such 'significantly higher' temperature to permit the
derivation methodology to be successful. It would be very useful for the reader to know what magnitude of higher temperature is necessary for the derivation methodology to provide valid wind results.

Figure 11 could be eliminated from the paper without the paper's impact/presentation being compromised.

Technical Corrections: The manuscript will benefit from English language editing.

---

## Author Comment (AC2) · 18 May 2017

Specific comments:

1.  *There are previously published VL1 winds derived following the quadrant sensor failure (Murphy et al., 1990). The manuscript does not provide any comparisons between those previous derived wind vectors and those derived using the method in this manuscript; such comparisons are warranted, as is some additional measurement-by-measurement comparison with the SANMET derived winds during the initial 45 sols (in addition to hourly averaged comparison, Figures 25 & 26). The impact of the manuscript will be strengthened with the inclusion of presentation of some derived point-by-point wind speed and direction, in addition to the hourly averaged presentation included in the current draft. Will the derived wind data set be made publicly available*

*in its entirety?*

Unfortunately a reasonable comparison between previous data and the data derived using the method described in the manuscript is currently not feasible, since we are not aware of the full point data of the previous reconstruction being publicly available.

The corrected version includes point-by-point wind speed and direction, in addition to the hourly averaged presentation.

The data is currently in an immature state and it will be made completely available later on, likely through the PDS-system. At present the data is available upon request, understanding that the data may be subject to changes.

2. *There is an apparent high-wind-speed bias that arises from the manuscript's methodology when compared to SANMET winds during sols 1-45. The manuscript does address this result but more attention is warranted to provide a more substantial basis for believing the derived wind speeds and directions.*

The reviewer is correct in pointing out the high-speed bias, arising from the manuscripts reconstruction method. However, the exact reason for this bias is still not understood well, and thus a fully satisfactory explanation is regrettably not possible at this time. We plan to further address the concern in a future publication, where a more detailed scientific analysis is performed.

3. *It would be helpful if Figure 3 included Sol 45 quadrant sensor signals to illustrate what a fully uncompromised sol's measurements exhibit. The presentation would benefit from displaying the nominal quadrant sensor signal during a complete sol which subsequently transitioned to the instrument behavior change evident during the first few hours of Sol 46 and Sol 47 which then became persistent during Sol 48.*

Figure 3 was edited to illustrate the nominal quadrant sensor signal during sol 45 as well the transition to the error state during sols 46 and 47.

4. *The substantial comparison within the paper of the newly derived wind vector results*

*with wind vectors derived applying the SANMET software to the degraded sensor signals is unwarranted. [Figures 24, 25, and 26.] The SANMET software was designed to operate with signals from fully functional instruments. There is no doubt that SANMET-derived wind speeds and directions from the compromised instruments will be flawed, and using such flawed results as a comparison with the newly derived wind speeds and directions does not itself provide validation of the newly derived results. Rather, it is better to compare the newly derived winds with anticipated environmental conditions and their presumed physical driving of the winds that were experienced. For instance, the winds experienced at VL1 during the two global scale dust storms that occurred during the first year of the mission, initiating at Ls ~205 and ~270 (sols ~210 and ~315), are theoretically expected to have exhibited a semi-diurnal rotation of the wind vector, which the derived winds from Murphy et al (1990) were in agreement with.*

*The hodograph figures in Murphy et al illustrate this wind vector rotation arising from amplified thermal tides. Also, Viking Lander 1 camera images of the landing site provided evidence of surface material motion believed to be due to wind stress, requiring fast wind speeds from a direction indicated by the direction of material motion (Sagan et al, JGR, 82, September 1977, 4430-4438; Moore, H., JGR, 90, November 1985, 163-174).*

The reviewer is correct pointing out that the SANMET-process for wind instruments is invalid after sol 45 of the mission, when the instruments were not fully operable. The article presents two validation methods for the wind reconstruction algorithm in Sec. 4. The first validation method is based on fully functional wind instruments and it is conducted with the data from sols 1-45. The second validation method is based on the expected slope-induced winds in the VL1 area.

The previous Figures 25 and 26 were replaced by the following two figures: The first Figure contains a comparison with the SANMET data from sol 15, when the VL1 wind instruments were fully functional. The second Figure contains the reconstructed winds from sol 1413. The sol 1413 has the same season as sol 15, but it is exactly two years

after sol 15. Therefore the winds from the both sols should exhibit similar behavior, which is discussed shortly in the text.

5. *In Section 3.1 the word 'segment' is invoked to describe each of the two time intervals (sols 46-377, sols 378-2245) during which the two specific wind derivation methodologies are implemented/applied. Since 'segment' is frequently used to describe a portion of a physical structure rather than a time interval, I suggest considering replacing 'segment' with a word that unambiguously indicates time, such as 'stage'. Stage 1 could correspond to the failed quadrant sensor during sols 46-377 while the wind sensor continued operating, while Stage 2 could correspond to the subsequent failed Wind Sensor element condition.*

The word 'stage' is better in this situation as it unambiguously indicates time. The word 'segment' was replaced by 'stage'.

6. *Since the concept of Nusselt number (introduced on Page 3) is very important to the paper's discussion of the wind sensor signals, I recommend providing a definition of Nusselt Number within the text.*

The definition of the Nusselt number was added to Sec. 2.1.

7. *In Section 2.4, final sentence of the 2nd paragraph, the statement '.. a significantly higher temperature than the ambient temperature' occurs, but there is no declaration as to the necessary magnitude of such 'significantly higher' temperature to permit the derivation methodology to be successful. It would be very useful for the reader to know what magnitude of higher temperature is necessary for the derivation methodology to provide valid wind results.*

The threshold voltage, when the quadrant sensor's thermocouples signals could be used for determining the wind directions, were added in Sec. 2.4.

8. *Figure 11 could be eliminated from the paper without the paper's impact/presentation being compromised.*

Figure 11 was not critical for the presentation of the paper, therefore Figure 11 was removed.

Technical corrections:

1. *The manuscript will benefit from English language editing.*

A comprehensive language editing has been performed to the manuscript.

Thank you very much for your comments.

**T. Kynkäänniemi et al.**
* * *
**Interactive comment on Geosci. Instrum. Method. Data Syst. Discuss., doi:10.5194/gi-2016-41, 2017.**